# Language-Guided Visual Prompt Compensation for Multi-Modal Remote Sensing Image Classification with Modality Absence

## ABSTRACT

Joint classification of multi-modal remote sensing images has achieved great success thanks to complementary advantages of multi-modal images. However, modality absence is a common dilemma in real world caused by imaging conditions, which leads to a breakdown of most classification methods that rely on complete modalities. Existing approaches either learn shared representations or train specific models for each absence case so that they commonly confront the difficulty of balancing the complementary advantages of the modalities and scalability of the absence case. In this paper, we propose a language-guided visual prompt compensation network (LVPCnet) to achieve joint classification in case of arbitrary modality absence using a unified model that simultaneously considers modality complementarity. It embeds missing modality-specific knowledge into visual prompts to guide the model in capturing complete modal information from available ones for classification. Specifically, a language-guided visual feature decoupling stage (LVFD-stage) is designed to extract shared and specific modal feature from multi-modal images, establishing a complementary representation model of complete modalities. Subsequently, an absence-aware visual prompt compensation stage (VPC-stage) is proposed to learn visual prompts containing missing modality-specific knowledge through cross-modal representation alignment, further guiding the complementary representation model to reconstruct modality-specific features for missing modalities from available ones based on the learned prompts. The proposed VPC-stage entails solely training visual prompts to perceive missing information without retraining the model, facilitating effective scalability to arbitrary modal missing scenarios. Systematic experiments conducted on three public datasets have validated the effectiveness of the proposed approach.

## KEYWORDS

Joint classification, Multi-modal, Modality absence, Language-visual model, Prompt learning

## 1 INTRODUCTION

Joint classification of multi-modal remote sensing images is an efficient technique that integrates information from various modalities to achieve precise classification of land unit elements[9]. It plays a crucial role in the earth observation tasks such as land analysis and utilization[12], urban planning and management[3], as well

*ACM MM, 2024, Melbourne, Australia*
© 2024 Copyright held by the owner/author(s). Publication rights licensed to ACM.
ACM ISBN 978-x-xxxx-xxxx-x/YY/MM
https://doi.org/10.1145/nnnnnnn.nnnnnnn

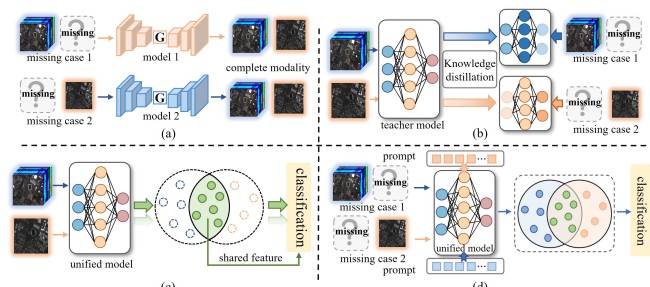

**Figure 1: Illustrations of different methods for addressing missing modalities. (a) Generation-based methods. (b) Transfer learning-based methods. (c) modality-shared latent space learning methods. (d)The proposed method.**

as environmental conservation and monitoring[28]. Recently, the researches of joint classification have shown great success, with their outstanding performance relying on the exploration of complementary advantage from complete modalities[7, 31, 42]. However, modality absence is a common dilemma[21, 38] in real-world scenarios due to sensor malfunctions or inconsistent satellite revisit period. This makes it challenging for traditional classification models to extract effective discriminative features from the limited modal data, resulting in a significant degradation in classification performance. Therefore, it becomes essential to develop joint classification method that can cope with modality absence.

The existing methods to address the issue of modality absence in multi-modal classification can be categorized into three types: generation-based[36, 46], transfer learning-based[33, 34] and modality-shared latent space learning methods[6, 10]. The generation-based methods restore missing modality images by synthesizing information from available modalities through generative network[2, 46]. Nevertheless, due to the instability of image generation, this may introduce considerable noise, which is harmful for classification[8]. Transfer learning-based methods typically transfer knowledge from a full-modal network to a network with modality absence through knowledge distillation[14], thereby optimizing the classification boundary with modality absence. Whereas it is challenging to guide the network with modality absence to inherit complete modal information owing to significant heterogeneity among remote sensing images of different modalities, which may lead to sub-optimal performance[35]. Although promising results can be obtained, these methods require training a specific model for each missing scenario[37], which undoubtedly introduces a significantly additional training parameters, severely limiting their scalability to application scenarios with arbitrary modality absence. To alleviate this limitation, modality-shared latent space learning methods aim to learn a unified model for various modal combinations[45]. It utilizes the latent commonality between modalities for classification, typically establishing shared subspace for all modalities and learning modality-invariant features to mitigate the influence of modality gap[6].

However, the discriminative ability in feature representation of such methods is limited since they solely focus on modality-shared features[19], neglecting modality-specific information, thus forfeiting the complementary advantages of multi-modality. These discussions motivated us to pose a research question: Can a unified model be constructed that simultaneously considers both modality-shared and modality-specific information, while remaining robust to arbitrary modality absence without incurring significantly additional training parameters?

To address the above issue, we draw inspiration from prompt learning. The essence of prompt learning is to design prompts for downstream tasks, guiding pre-trained models to perform the anticipant tasks without modifying itself, where knowledge about the task is embedded as prompts in input tokens to help network understand the meaning of task. Inspired by this, we propose a language-guided visual prompt compensation network (LVPCnet) to achieve joint classification of multi-modal remote sensing images in case of modality absence. In this designed framework, the classification model can be guided to capture modality-specific information of missing modalities from available ones by learning a visual prompt that can perceive the missing modality knowledge, enabling the acquisition of complete modal information for classification. Concretely, it is achieved by a two-stage training process: language-driven visual feature decoupling stage (LVFD-stage) and absence-aware visual prompt compensation stage (VPC-Stage). The LVFD-stage decomposes multi-modal images into modal-shared and modality-specific representations through a shared encoder and multiple specific encoders, establishing a complementary feature representation framework. Unlike common decomposition methods, we employ modality attribute-associated language priors to guide the decoupling of multi-modal visual features under multi-dimensional visual-language alignment constraints. This approach leverages the rich semantic information provided by language to help the visual system better understand and interpret modal content. The proposed VPC-stage takes available modalities as input to the pre-trained feature representation framework of the LVFD-stage, and integrates visual prompts with specific encoders for missing modalities and employ cross-modal representation alignment. This allows visual prompts to learn specific knowledge about missing modalities, thereby guiding these specific encoders to extract specific features of missing modalities from available ones.

To summarize, the contributions of this work are as follows:

- We propose a unified model LVPCnet for joint classification with arbitrary modal absence, which incorporates the modality complementarity through reconstructing the specific feature of missing modalities by learning visual prompts capable of perceiving missing modality-specific knowledge.

- We design an language-driven visual feature decoupling stage (LVFD-stage) for multimodal image decoupling, where language priors are utilized to explicitly guide the model to capture modality-specific knowledge, facilitating subsequent visual prompts to adeptly acquire the specific knowledge associated with the absent modality.

- We design absence-aware visual prompts for guiding the compensation of missing modality-specific features from the available ones, a process that only requires training the

prompts without modifying original model, facilitating extension to arbitrary missing scenarios.

## 2 RELATED WORK

### 2.1 Modality Absence in Multi-modal Learning

The issue of modality absence is common in multi-modal learning due to the limitations of imaging conditions, and several studies have emerged to provide solutions for overcoming modality absence[18, 29, 39]. Ma et al. [22] proposed the SMIL model, which applied a Bayesian meta-learning framework to learn the weighted sum of modal priors from complete modalities to reconstruct the features of missing modalities. Pande et al.[23] introduce an adversarial training-driven hallucination architecture that employs a cross-modal hallucination module based on C-GAN to generate discriminative features related to the missing modality from available modalities. MMIN[41] leverages a cascaded residual autoencoder for cross-modal imagination to learn joint multi-modal representations for classification. Wang et al.[30] proposed a learnable cross-modal knowledge distillation model for adaptive recognition of significant modalities and knowledge from them to assist other modalities in addressing modality deficiency from a cross-modal perspective.

### 2.2 Prompt Learning

The concept of prompt learning was initially introduced in the field of natural language processing[43]. It adapts to various downstream tasks by modifying prompt instead of adjusting the pre-trained language model. Presently, prompt learning have been incorporated into tasks related to computer vision[15, 25]. Coop[44] utilized learnable vectors in a continuous space to represent the prompt of context, while maintained fixed parameters for the entire CLIP pre-trained model. MaPLe[16] employed interactive prompt in both visual and language domains simultaneously to enhance the consistency of representations between vision and language. PromptFuse[20] utilized prompt vectors to align modalities, adapting to downstream multi-modal tasks in a modular and parameter-efficient manner. These studies suggested that prompt learning can effectively adapt to various tasks in different input scenarios. This provides us with an idea of integrating prompt learning into multi-modal learning, and prompt learning can be applied to adapt to multi-modal learning in case of missing modalities.

## 3 METHOD

### 3.1 Overview

Given a multi-modal images dataset with $m$ modalities, we assume $m = 2$ for simplicity and without losing generality. It is denoted by $D = \{\mathbf{X}^{m_i}, \mathbf{X}^{m_j}, y\}$ where $\mathbf{X}^{m_i}$ and $\mathbf{X}^{m_j}$ are the images of modality $m_i$ and $m_j$, and $y$ is the category labels. Then, an incomplete modality case can be represented as $D^{m_i} = \{\mathbf{X}^{m_i}, y\}$ or $D^{m_j} = \{\mathbf{X}^{m_j}, y\}$. The proposed LVPCnet aims to accurately predict category labels $y$ from either complete or incomplete modal images during inference by fully leveraging the information from complete modalities during training. Considering that the missing modality of each input data cannot be predicted in advance in real-world scenarios, training a separate model for each missing scenario would undoubtedly

**Figure 2: Overall architecture of the proposed LVPCnet. The method consists of two stage:1) language-driven visual feature decoupling stage for extraction of shared and specific visual features. 2) absence-aware visual prompt compensation stage for the reconstruction of missing modality-specific features.**

introduce a large number of additional training parameters. To alleviate this issue, we propose a unified classification model to address arbitrary modality absence, which is achieved by learning visual prompts capable of perceiving missing modality knowledge. The learned prompts can guide the classification model to capture specific modality information of the missing modality from available ones, thus obtaining complete modal information for classification.

The overview of our framework is depicted in Figure 2, a two-stage framework is employed to direct prompts for the compensation of missing modality. To make visual prompts focus on learning modality-specific information, we propose a visual feature decoupling stage LVFD-stage to separate shared and specific information in multi-modal images. Considering the complexity of image distribution which makes it difficult for the prompts to fully learn the modality-specific representations, we utilize language priors to drive the decoupling representation of multi-modal images. The superiority of this strategy to explicitly guide the representation of image content instead of complex distribution can be attributed to the ability of language in capturing abstract concepts and descriptions of relationships in the images. Subsequently, an absence-aware visual prompts compensation stage VPC-stage is proposed to utilize visual prompts to guide the reconstruction of missing modality-specific features to complete the modal information.

Specifically, LVFD-stage takes multi-modal remote sensing images $D$ as input. Under the constraint of multi-dimensional language-visual contrastive alignment, each modality image is separately input into shared and respective modality-specific visual encoders to obtain modality-shared and modality-specific features, which are represented as $\{c_{m_i}, c_{m_j}\}$ and $\{s_{m_i}, s_{m_j}\}$, respectively. To ensure the completeness of complementary feature representation in the absence of certain modalities, particular attention should be given to compensating for specific features $s_{m_i}$ or $s_{m_j}$. To this end, the VPC-stage takes available modalities $m_i$ (or $m_j$) as input, while integrating visual prompts into the visual encoders specifically to missing modalities, aiming to learn the mapping from modality $m_i$ to $m_j$ in the modality-specific latent feature space by optimizing the prompts. This enables the derivation of the specific representation $s'_{m_j}$ (or $s'_{m_i}$) for modality $m_j$ (or $m_i$) based on $m_i$ (or $m_j$) when $m_j$ (or $m_i$) is missing. In this case, complete complementary features can be obtainable for classification, as follows:

$$s'_{m_j} = E^{m_j}_{sp}(\{\mathbf{X}^{m_i}; \mathbf{P}\})$$
$$\hat{y} = CLS(c_{m_i}, s_{m_i}, s'_{m_j}) \qquad (1)$$

where $E^{m_j}_{sp}$ denotes the specific encoder of $m_j$ modality, $\mathbf{P}$ represents the absence-aware visual prompt specific to $m_j$ modality and $\hat{y}$ denotes the predicted category labels.

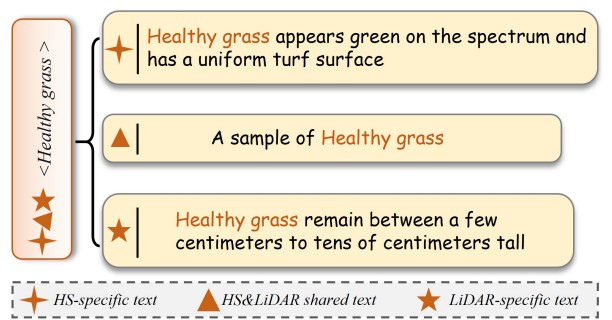

**Figure 3: Examples of language descriptions for HSI and LiDAR**

## 3.2 Language-Driven Visual Feature Decoupling

Due to the presence of feature information overlap in multi-modal features, directly recovering the features of missing modalities may lead to a greater emphasis on recovering over-lapping parts, potentially weakening the reconstruction effect of specific information. Therefore, it is necessary to decompose the complementary features of the multi-model data into modality-shared and modality-specific features, which facilitates targeted compensation for the missing specific features in subsequent steps. The traditional decoupling methods often merely separate features without providing guidance for expressing the content of modalities. Meanwhile, language can provide rich semantic information to assist the visual system to better understand and interpret images contents. To this end, language prior knowledge is introduced to guide the decoupling of multi-modal visual features. It is extracted from the pre-trained large language model, which is based on the land cover diversity of the multi-modal images, and can be decomposed into shared and specific language priors. Shared and specific language features are extracted from the priors through a language encoder, and the visual features of different modalities are aligned with their corresponding language features through contrastive learning to achieve feature decoupling. In order to extract effective language priors and guide visual representations, both language feature establishment and multi-dimensional visual-language alignment aspects are comprehensively considered.

### 3.2.1 Language Feature Establishment.
It is well known that language, as a comprehensive descriptor of land cover information, can reflect the representation forms of land cover characteristics in different visual modalities. Based on this, we establish language priors for each category of land cover, encompassing both shared and modality-specific information, which is achieved by providing modal attribute-related guidelines to a large language generation model. We consider the fact that different modal images share the same information in the semantic space as a basis for describing modality-shared attributes, utilizing the template "$\langle class; Name \rangle$" to expand complete shared language descriptions. For specific aspects, it depends on the information of land cover reflected by each modality. For example, for two data modalities, hyperspectral image (HSI) can better reflect spectral information, so language descriptions for land cover features regarding color and material can be derived from HSI. On the other hand, LiDAR can efficiently and accurately obtain elevation information of the ground compared

to HSI. Therefore, height information serves as specific language descriptions for LiDAR. An example is shown in Figure 3.

After acquiring the language descriptions, a text encoder is employed to extract shared and specific language features from the corresponding language descriptions, which is constructed through the pre-trained transformer architecture of CLIP[24] that is widely used in language models. It utilizes lower-cased byte pair encoding (BPE) to obtain tokenized representations of the text, which are then passed through the the fixed-parameter transformer for encoding to extract modality-specific features denoted as $\mathbf{F}_l^{sm_i}, \mathbf{F}_l^{sm_j}$ and modality-shared semantic features represented as $\mathbf{F}_l^c$.

### 3.2.2 Multi-Dimensional Visual-Language Alignment.
For the input multi-modal images $\mathbf{X}^{m_i}$ and $\mathbf{X}^{m_j}$, encoders are designed for feature embedding. Specifically, two independently optimized specific encoders $E_{sp}^{m_i}(\cdot)$ and $E_{sp}^{m_j}(\cdot)$ are designed for extracting specific features, and the other parameter-shared shared encoder $E_{sh}(\cdot)$ for extracting shared features across different modalities. It can be formulated as follows:

$$\mathbf{F}_v^{cm_i} = E_{sh}(\mathbf{X}^{m_i}), \ \mathbf{F}_v^{cm_j} = E_{sh}(\mathbf{X}^{m_j})$$
$$\mathbf{F}_v^{sm_i} = E_{sp}^{m_i}(\mathbf{X}^{m_i}), \ \mathbf{F}_v^{sm_j} = E_{sp}^{m_j}(\mathbf{X}^{m_j}) \tag{2}$$

where $\mathbf{F}_v^{cm_i}$ and $\mathbf{F}_v^{cm_j}$ represent the extracted shared features from the respective modalities, and $\mathbf{F}_v^{sm_i}$ and $\mathbf{F}_v^{sm_j}$ denote the specific features from each modality, respectively. Here, $E_{sh}(\cdot), E_{sp}^{m_i}(\cdot)$ and $E_{sp}^{m_j}(\cdot)$ follow the same structure as the ViT, with an additional MLP for projecting the features into the common space.

In order to optimize shared and specific visual encoders for more comprehensive extraction of decoupled modality complementary information, a multi-dimensional visual-language alignment strategy has been proposed, which is implemented by aligning language-shared and language-specific features with modality-shared and modality-specific visual features respectively through contrastive learning between images and language pairs. Unlike traditional contrastive learning between individual language-image pairs, we assign the same language to all images of the same category, whether it's a shared or specific description. Treating all visual features of the same category with the same language as positive samples aims to maximize the similarity between their feature vectors, effectively reducing intra-class variance while widening the inter-class gap. For the shared visual features of modalities $m_i$ and $m_j$, we align both of them with the shared language features, the loss takes the following form:

$$\mathcal{L}_{shared} = \mathcal{L}_{(\mathbf{F}_l^c, \mathbf{F}_v^{cm_i})} + \mathcal{L}_{(\mathbf{F}_l^c, \mathbf{F}_v^{cm_j})} \tag{3}$$

Where $\mathcal{L}_{(\mathbf{F}_l^c, \mathbf{F}_v^{cm_i})}$ and $\mathcal{L}_{(\mathbf{F}_l^c, \mathbf{F}_v^{cm_j})}$ denote the align loss between the shared language and the images of modalities $m_i$ and $m_j$, respectively. Taking $\mathcal{L}_{(\mathbf{F}_l^c, \mathbf{F}_v^{cm_i})}$ as an example, and $\mathcal{L}_{(\mathbf{F}_l^c, \mathbf{F}_v^{cm_j})}$ is computed in the same way. The image-to-text and text-to-image alignment losses are computed as:

$$\mathcal{L}_{(\mathbf{F}_l^c, \mathbf{F}_v^{cm_i})} = -\sum_{n=0}^{N} \frac{1}{|P(n)|} \left( \sum_{p \in P_l(n)} \log \frac{\exp([(\mathbf{F}_v^{cm_i})_n]^T [(\mathbf{F}_l^c)_p]^+ / \tau)}{\sum_{a \in A_l(n)} \exp([(\mathbf{F}_v^{cm_i})_n]^T [(\mathbf{F}_l^c)_a]^- / \tau)} \right.$$
$$\left. + \sum_{p \in P_v(n)} \log \frac{\exp([(\mathbf{F}_l^c)_n]^T [(\mathbf{F}_v^{cm_i})_p]^+ / \tau)}{\sum_{a \in A_v(n)} \exp([(\mathbf{F}_l^c)_n]^T [(\mathbf{F}_v^{cm_i})_a]^- / \tau)} \right) \tag{4}$$

here, for each embedding feature $\mathbf{F}_v^{cm_i}$ and $\mathbf{F}_l^c$ in minibatch, $P_v(n)$ and $A_v(n)$ are the sets of all positive and negative samples of visual features respectively, and $P_v(n)$ and $A_v(n)$ are their cardinality. Similarly, $P_l(n)$ and $A_l(n)$ represent the positive and negative sample sets for language features. The language and visual features belonging to the same category are put into $P_l(n)$ and $P_v(n)$, and the out-of-class features are put into $A_l(n)$ and $A_v(n)$. $\tau$ is a scalar temperature parameter. $\mathcal{L}_{spe\_k}$ represents the alignment loss of the $k$-th modality-specific features, computed similar to $\mathcal{L}_{(\mathbf{F}_l^c, \mathbf{F}_v^{cm_i})}$.

## 3.3 Absence-Aware Visual Prompt Compensation

VPC-stage is designed to recover specific information of missing modalities. To minimize the introduction of additional parameters, we are inspired by the idea of prompt learning and design a visual prompt to learn specific knowledge of the missing modality. This prompt is then used to guide the model in the LVFD-stage stage to extract specific features of the missing modality from available ones. We achieve this by integrating visual prompts into specific encoders of the missing modality, taking available modalities as input to these encoders, and aligning the output features with the specific language features of the missing modality. During training, the only trainable parameters are the absence-aware visual prompts used to learn the missing modality features. We illustrate the compensation of specific features for modality $m_i$ with missing modality $m_i$ as an example.

A dimension matching operation is employed to unify the dimensions of the available modal input $\mathbf{X}^{m_j} \in \mathbb{R}^{H \times W \times C_2}$ with the missing modal $\mathbf{X}^{m_i} \in \mathbb{R}^{H \times W \times C_1}$, ensuring that it meets the input requirements of the specific encoder for modality $m_i$. The dimension matching is accomplished through a convolutional layer, which can be represented as:

$$\hat{\mathbf{X}}^{m_j} = conv(\mathbf{X}^{m_j}) \tag{5}$$

where $\hat{\mathbf{X}}^{m_j} \in \mathbb{R}^{H \times W \times C_1}$ represents the output after dimension matching. Then the $\hat{\mathbf{X}}^{m_j}$ is divided into $n$ patches $\{\mathbf{I}_q \in \mathbb{R}^{p \times p \times C_1} | 1 \le q \le n\}$, $p \times p$ denotes the size of patches. Each patch is projected into $d$-dimensional latent space, as follows

$$e_0^q = Proj(\mathbf{I}_q), e_0^q \in \mathbb{R}^d, 1 \le q \le n \tag{6}$$

where $Proj(\cdot)$ represents the projection operation. To combine the available modality $m_j$ with visual prompts to compensate for the specific features of modality $m_i$ we effectively adapt the specific visual encoder of modality $m_i$ with modality $m_j$ through visual prompts. As mentioned before, the visual encoder is based on the ViT structure, which generally consists of a cascade of $N$ encoder layers (here $N = 4$). We denote the patch embedding features of layer $l$ as $\mathbf{E}_l = \{e_l^q \in \mathbb{R}^d | 1 \le l \le N, 1 \le q \le n\}$, where $\mathbf{E}_l \in \mathbb{R}^{n \times d}$. Then absence-aware prompts are introduced into the input space of each Transformer layer, which is attached to the embedding feature together with an extra learnable classification token to form an extension feature. For $l$-th layer $L_l$, the prompts are denoted as $\mathbf{P}_i \in \mathbb{R}^{l_p \times d}$ and randomly initialized, where $l_p$ is the prompt length. Finally, its output feature is denoted as:

$$\widehat{\mathbf{F}}_v^{m_i} = L_N(\cdots L_l(\cdots L_0(x_0^{cls}; \mathbf{P}_0; \mathbf{E}_0) \cdots ; \mathbf{P}_l) \cdots ; \mathbf{P}_N) \tag{7}$$

where $x_0^{cls} \in \mathbb{R}^d$ denotes classification token, $(\cdot; \cdot)$ represents the concatenation operations along the dimension of sequence length. In order to enable the visual prompts to thoroughly learn the missing modality-specific information, we align the output features $\widehat{\mathbf{F}}_v^{m_i}$ with the specific language features of the missing modality $m_i$, which is formulated as:

$$\mathcal{L}_{cross} = - \sum_{n=0}^{N} \frac{1}{|P(n)|} \Big( \sum_{p \in P_l(n)} \log \frac{\exp([(\widehat{\mathbf{F}}_v^{m_i})_n]^T [(\mathbf{F}_l^{sm_i})_p]^+ / \tau)}{\sum_{a \in A_l(n)} \exp([(\widehat{\mathbf{F}}_v^{m_i})_n]^T [(\mathbf{F}_l^{sm_i})_a]^- / \tau)}$$
$$+ \sum_{p \in P_v(n)} \log \frac{\exp([(\mathbf{F}_l^{sm_i})_n]^T [(\widehat{\mathbf{F}}_v^{m_i})_p]^+ / \tau)}{\sum_{a \in A_v(n)} \exp([(\mathbf{F}_l^{sm_i})_n]^T [(\widehat{\mathbf{F}}_v^{m_i})_a]^- / \tau)} \Big) \tag{8}$$

where $\mathbf{F}_l^{sm_i}$ represents the specific language features, $\widehat{\mathbf{F}}_v^{m_i}$ is the compensated visual features.

## 3.4 Training Objective

**Stage1: Modality Feature Decoupling**   During the modality feature decoupling, a joint optimization objective is defined to extract complementary decomposed features from multi-modal data, which contains a combination of multiple contrastive loss and joint classification loss:

$$\mathcal{L}_{stage1} = \lambda_1 \mathcal{L}_{con} + \lambda_2 \mathcal{L}_{cls} \tag{9}$$

where the hyperparameter $\lambda_1$, $\lambda_2$ control the balance of multiple losses. The multiple contrastive loss is composed of both shared feature alignment loss and all modality-specific feature alignment losses:

$$\mathcal{L}_{con} = \mathcal{L}_{shared} + \sum_{k}^{m} \mathcal{L}_{spe\_k} \tag{10}$$

here $\mathcal{L}_{shared}$ and $\mathcal{L}_{spe\_k}$ denotes the alignment loss for shared features and specific features of modality $m_k$, respectively.

The classification loss $\mathcal{L}_{cls}$ further optimizes the visual encoder, enhancing the extraction of more discriminative features. The computation formula is as follows:

$$\mathcal{L}_{cls} = - \sum_{n=1}^{N} y_n \log(\hat{y}_n) \tag{11}$$

where $N$ is the number of classes.

**Stage2: Compensation of Specific Features**   This stage constructs cross-modal visual-language alignment loss $\mathcal{L}_{cross}$ and classification loss $\mathcal{L}_{cls}$ for feature compensation and classification,

$$\mathcal{L}_{stage2} = \lambda_3 \mathcal{L}_{cross} + \lambda_4 \mathcal{L}_{cls} \tag{12}$$

here, $\mathcal{L}_{cross}$ is the loss described in Section 3.3, $\lambda_3$ and $\lambda_4$ are hyperparameters.

## 4 EXPERIMENTS

### 4.1 Datasets Description

We conduct experiments on three publicly multi-modal datasets for performance evaluation. A brief description of these three datasets is as follows:

1) Houston2013[4]   This dataset is part of the 2013 IEEE GRSS Data Fusion Competition and contains hyperspectral (HS) and Li-DAR images depicting 15029 labeled samples of 15 categories.

2) Trento[26]   The dataset was taken in a rural area south of Trento and consists of HS and LiDAR data, with a total of 30214 labeled samples in 6 land cover categories.

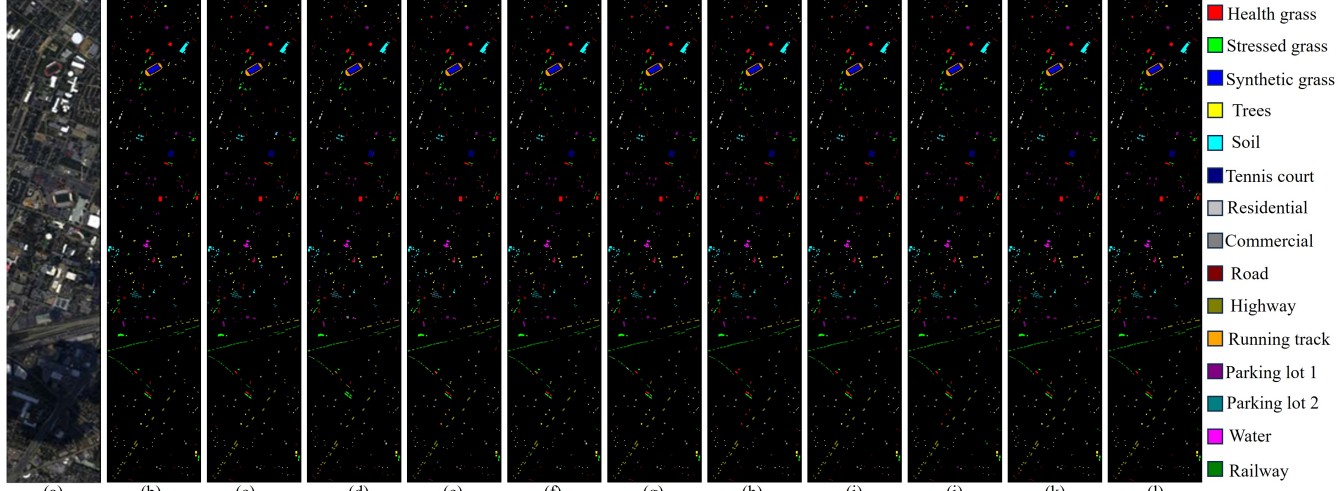

**Figure 4: Classification maps of the Houston2013 dataset. (a)Houston2013 dataset (HS). (b)Ground truth. (c)HS-net. (d)MDL-RS. (e) Cospace. (f)MSH-Net. (g)LVPCnet(HS). (h)MFT-Net. (i)GLT-Net. (j)Sal2RN. (k) HCT-Net. (l)LVPCnet(HS and LiDAR)**

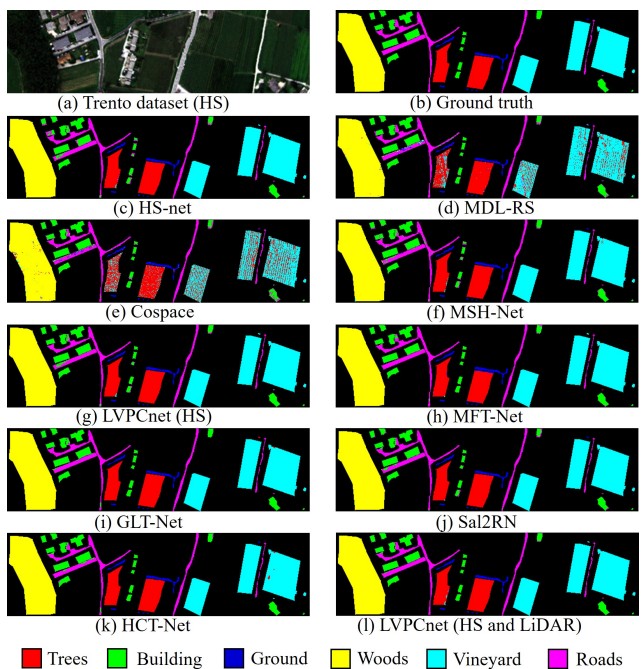

**Figure 5: Classification maps of the Trento dataset.**

3) Augsburg[1]   This dataset originates from Augsburg, Germany, comprising three modalities of data: HS, SAR, and LiDAR. It encompasses 7 categories with a total of 78294 labeled samples.

## 4.2 Experiments Setup

*4.2.1 Evaluation Metrics and Implementation Details.* Three evaluation metrics[32] are employed for quantitative analysis, including overall accuracy (OA), average accuracy (AA), and kappa coefficient.

The proposed method is implemented on the PyTorch platform and trained on two NVIDIA GeForce 3090 GPUs using the Adam optimizer. The model is trained for 500 epochs in the LVFD-stage, followed by 300 epochs in the VPC-stage. The batch size is set as 1024. The learning rate is initially set to 1e-3, and updated by a CosineAnnealingLR strategy. All the comparison methods selected 40 samples for training.

*4.2.2 Competing Methods.* To demonstrate the effectiveness of the proposed method in the joint classification of multi-modal remote sensing images in case of modality absence, we set up three different experimental configurations: 1) Training and testing occur within single modality in the proposed method, with each model named after the modality it utilizes. For instance, "HS-net" refers to a model trained with HS images, while "LiDAR-net" indicates a model trained with LiDAR images. 2) State-of-the-art methods for joint classification in case of modality absence, including Cospace[13], MDL-RS[11] and MSH-Net[35]. 3) State-of-the-art methods for joint classification with complete modalities: MFT[27], Sal2RN[17], HCT[40] and GLT-Net[5].

## 4.3 Comparison with State-of-the-Art Methods

*4.3.1 Results and Analysis on HS-LiDAR.* The left and middle parts of Table 1 show the performance comparison of OA, AA, and kappa on Houston2013 and Trento datasets under three types of comparison methods, respectively. Figures 4 and 5 show the classification maps of the comparison algorithms considered in the Houston and trento datasets, respectively. First, for scenarios without modality absence, it is evident that the model HS-LiDAR-Net trained with multi-modality outperform the HS-Net and LiDAR-Net trained with only uni-modality, which clearly demonstrates the complementary advantages of multi-modality. In the absence of HS or LiDAR modalities, the OA of HS-LiDAR-Net drops by 10.37% and 1.54% on the Trento dataset. On the Houston2013 dataset, the decrease is more pronounced, with declines of 16.81% and 1.97%. This indicates the ineffectiveness of applying traditional multi-modal models to the case of modal incompleteness. In contrast, the proposed LVPCnet

**Table 1: Classification accuracy of different methods on Trento, Houston and Augsburg Datasets. "W/o" denotes the missing modality in inference. "LiDAR/SAR-Net" indicates models trained and tested with only LiDAR or SAR images.**

| Method | Houston2013 | | | | | Trento | | | | | Augsburg | | | | |
|---|---|---|---|---|---|---|---|---|---|---|---|---|---|---|---|
| | Training Modalities | Testing Modalities | OA(%) | AA(%) | Kappa | Training Modalities | Testing Modalities | OA(%) | AA(%) | Kappa | Training Modalities | Testing Modalities | OA(%) | AA(%) | Kappa |
| | single modality | | | | | single modality | | | | | single modality | | | | |
| HS-Net | HS | HS | 97.01 | 97.46 | 96.78 | HS | HS | 98.31 | 96.99 | 97.76 | HS | HS | 90.44 | 87.95 | 86.73 |
| LiDAR/SAR-Net | LiDAR | LiDAR | 82.26 | 83.08 | 80.84 | LiDAR | LiDAR | 90.41 | 85.57 | 87.36 | SAR | SAR | 84.95 | 74.61 | 79.34 |
| | W/o HS modality | | | | | W/o HS modality | | | | | W/o HS modality | | | | |
| HS-LiDAR/SAR-Net | HS, LiDAR | LiDAR | 81.67 | 83.68 | 80.22 | HS, LiDAR | LiDAR | 89.20 | 87.77 | 85.86 | HS, SAR | SAR | 83.64 | 77.77 | 77.80 |
| Cospace | HS, LiDAR | LiDAR | 34.69 | 36.75 | 29.80 | HS, LiDAR | LiDAR | 73.87 | 77.50 | 66.18 | HS, SAR | SAR | 35.03 | 35.17 | 23.26 |
| MDL-RS | HS, LiDAR | LiDAR | 73.51 | 76.64 | 71.51 | HS, LiDAR | LiDAR | 65.09 | 70.05 | 57.00 | HS, SAR | SAR | 52.87 | 54.97 | 42.78 |
| MSH-Net | HS, LiDAR | LiDAR | 61.09 | 61.19 | 58.13 | HS, LiDAR | LiDAR | 91.72 | 89.59 | 89.07 | HS, SAR | SAR | 61.93 | 56.32 | 52.01 |
| **LVPCnet (Ours)** | HS, LiDAR | LiDAR | **85.1** | **86.75** | **83.92** | HS, LiDAR | LiDAR | **94.99** | **93.87** | **93.38** | HS, SAR | SAR | **86.37** | **79.26** | **81.27** |
| | W/o LiDAR modality | | | | | W/o LiDAR modality | | | | | W/o SAR modality | | | | |
| HS-LiDAR/SAR-Net | HS, LiDAR | HS | 96.51 | 96.79 | 96.23 | HS, LiDAR | HS | 98.03 | 96.66 | 97.37 | HS, SAR | HS | 89.10 | 86.13 | 84.96 |
| Cospace | HS, LiDAR | HS | 87.24 | 87.44 | 86.20 | HS, LiDAR | HS | 85.25 | 88.46 | 80.57 | HS, SAR | HS | 58.01 | 59.82 | 47.88 |
| MDL-RS | HS, LiDAR | HS | 85.90 | 86.93 | 84.77 | HS, LiDAR | HS | 91.47 | 92.81 | 88.76 | HS, SAR | HS | 57.13 | 63.15 | 47.28 |
| MSH-Net | HS, LiDAR | HS | 96.33 | 96.91 | 96.04 | HS, LiDAR | HS | 98.59 | 97.79 | 98.12 | HS, SAR | HS | 87.31 | 79.06 | 82.45 |
| **LVPCnet (Ours)** | HS, LiDAR | HS | **98.05** | **98.22** | **97.90** | HS, LiDAR | HS | **99.07** | **98.24** | **98.77** | HS, SAR | HS | **91.01** | **87.08** | **87.43** |
| | complete modalties | | | | | complete modalties | | | | | complete modalities | | | | |
| MFT | HS, LiDAR | HS, LiDAR | 96.14 | 96.73 | 95.83 | HS, LiDAR | HS, LiDAR | 99.16 | 98.89 | 98.52 | HS, SAR | HS, SAR | 86.36 | 75.90 | 81.07 |
| Sal2RN | HS, LiDAR | HS, LiDAR | 97.34 | 97.75 | 97.12 | HS, LiDAR | HS, LiDAR | 99.19 | 98.66 | 98.91 | HS, SAR | HS, SAR | 91.62 | 81.80 | 88.25 |
| HCT | HS, LiDAR | HS, LiDAR | 96.80 | 97.41 | 96.54 | HS, LiDAR | HS, LiDAR | 99.22 | 98.90 | 98.96 | HS, SAR | HS, SAR | 88.68 | 80.93 | 84.25 |
| GLT-Net | HS, LiDAR | HS, LiDAR | 98.24 | 98.42 | 98.10 | HS, LiDAR | HS, LiDAR | 99.46 | 98.92 | 99.28 | HS, SAR | HS, SAR | 90.75 | 77.24 | 86.95 |
| **LVPCnet (Ours)** | HS, LiDAR | HS, LiDAR | **98.48** | **98.72** | **98.37** | HS, LiDAR | HS, LiDAR | **99.57** | **99.12** | **99.43** | HS, SAR | HS, SAR | **92.94** | **86.8** | **89.11** |

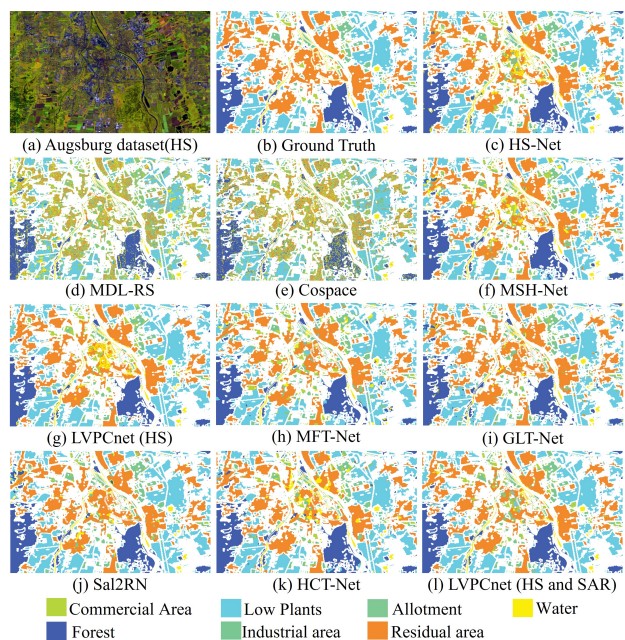

(a) Augsburg dataset(HS)    (b) Ground Truth    (c) HS-Net

(d) MDL-RS    (e) Cospace    (f) MSH-Net

(g) LVPCnet (HS)    (h) MFT-Net    (i) GLT-Net

(j) Sal2RN    (k) HCT-Net    (l) LVPCnet (HS and SAR)

■ Commercial Area  ■ Low Plants  ■ Allotment  ■ Water
■ Forest  ■ Industrial area  ■ Residual area

**Figure 6: Classification maps of the Augsburg dataset.**

significantly addresses this issue and yields superior performance to uni-modal models. Specifically, the OA of LVPCnet when LiDAR is missing outperforms the model trained with single modality by 0.76% and 1.04% on the Trento and Houston2013 datasets, respectively. Moreover, OA improved by 4.58% and 2.84% with the absence of HS images. Additionally, The LVPCnet outperforms the Cospace, MDL-RS and MSH-Net by 21.12%, 29.9% and 3.27% in terms of OA

respectively when HS is missing and by 13.82%, 7.6% and 0.48% respectively when LiDAR is missing on the Trento dataset. Similarly, it also performs better on the Houston dataset. This indicates that the proposed method effectively utilizes language priors to extract and compensate for more discriminative specific features of the missing modality compared to other methods.

*4.3.2 Results and Analysis on HS-SAR.* We conduct experiments for HS and SAR modalities on the Augsburg dataset to further evaluate the generalization performance of LVPCnet. As shown in the right part of Table1, the LVPCnet demonstrates superior performance compared to joint classification with complete modalities. Moreover, it even outperforms partially multi-modal fusion classification models when certain modalities are missing. The proposed method with absence of SAR images achieves improvements of 4.65%, 2.33% and 0.26% compared to the multi-modal joint classification method MFT, HCT and GLT-Net. This indicates that the proposed method not only deal with missing modality but also demonstrates significant potential in joint classification. Figure 6 shows the classification maps.

*4.3.3 Results and Analysis on HS, LiDAR and SAR.* We conduct experiments on modal combinations of HS, LiDAR, and SAR images to evaluate the scalability of our method in the presence of multi-modal absence. As shown in Table 2, we can observe that the proposed LVPCnet performs better than the conventional multi-modal model under the setting of missing modalities, and also exceeds the uni-modal model in Table 1 during single-modal testing, which suggests that LVPCnet has superior robustness even in the case of missing multiple modalities.

**Table 2: Performance on Augsburg dataset with HS, LiDAR and SAR. The Baseline means classification results of the conventional multi-modal model with missing modalities.**

| Method | Training Modalities | Testing Modalities | OA(%) | AA(%) | Kappa |
|---|---|---|---|---|---|
| Baseline | HS,LiDAR,SAR | HS | 89.58 | 84.54 | 85.49 |
| LVPCnet | HS,LiDAR,SAR | HS | **91.17** | **86.59** | **87.66** |
| Baseline | HS,LiDAR,SAR | LiDAR | 59.95 | 66.41 | 49.45 |
| LVPCnet | HS,LiDAR,SAR | LiDAR | **60.5** | 66.37 | **50.01** |
| Baseline | HS,LiDAR,SAR | SAR | 85.81 | 78.46 | 80.54 |
| LVPCnet | HS,LiDAR,SAR | SAR | **86.42** | 78.22 | **81.26** |
| Baseline | HS,LiDAR,SAR | HS,LiDAR | 89.82 | 85.37 | 85.8 |
| LVPCnet | HS,LiDAR,SAR | HS,LiDAR | **91.69** | **86.57** | **88.36** |
| Baseline | HS,LiDAR,SAR | HS,SAR | 91.61 | 85.44 | 88.27 |
| LVPCnet | HS,LiDAR,SAR | HS,SAR | **92.35** | **87.59** | **89.25** |
| Baseline | HS,LiDAR,SAR | LiDAR,SAR | 85.87 | 82.77 | 80.66 |
| LVPCnet | HS,LiDAR,SAR | LiDAR,SAR | **87.08** | **79.63** | **82.03** |

**Table 3: Results of ablation experiments on the effectiveness of visual prompts.**

| dataset | Method | Trainable parameters(K) | W/o LiDAR | | | W/o HS | | |
|---|---|---|---|---|---|---|---|---|
| | | | OA(%) | AA(%) | Kappa | OA(%) | AA(%) | Kappa |
| Trento | vp1 | - | 98.03 | 96.66 | 97.37 | 89.2 | 87.77 | 85.86 |
| | vp2 | 171584 | 99.32 | 98.24 | 99.10 | 95.12 | 93.59 | 93.6 |
| | Ours | 1024 | 99.07 | 98.24 | 98.77 | 94.99 | 93.87 | 93.38 |
| Augsburg | vp1 | - | 89.1 | 86.13 | 84.96 | 83.64 | 77.77 | 77.8 |
| | vp2 | 171584 | 91.5 | 86.63 | 88.07 | 86.46 | 74.76 | 81.28 |
| | Ours | 1024 | 91.01 | 87.08 | 87.43 | 86.37 | 79.26 | 81.27 |

details through feature decoupling. To validate its effectiveness, a variant named as 'vf', is designed to directly extract features from multi-modal data without feature decoupling. As shown in the results from Figure 7, LVPCnet achieves a more satisfactory performance than vf, which proves the effectiveness of feature decoupling.

*4.4.3 Effectiveness of Visual Prompts.* This paper enhances the learning of specific features from missing modalities by introducing visual prompts, without the need to modify the model or introduce additional networks. To validate the effectiveness and efficiency of visual prompts, we design two variants (named 'vp1' and 'vp2') to compare the performance of the proposed method. vp1 remove visual prompts and vp2 replaces visual prompts with reconstruction network. The comparison results are shown in Table 3, LVPCnet outperforms vp1 by a wide margin. As for vp2, there is a slight improvement in OA, but the accuracy improvement remained within 0.5%. The reason for this phenomenon is that the reconstruction network needs to retrain a network for each missing modality, introducing abundant training parameters as shown in Table 3. Comparatively, the proposed method only trains the prompts without retraining the original network, and the parameter count of the prompts was only 0.6% of the reconstruction network. Therefore, visual prompts are trained for each missing modality significantly reduces computational complexity compared to a reconstruction network.

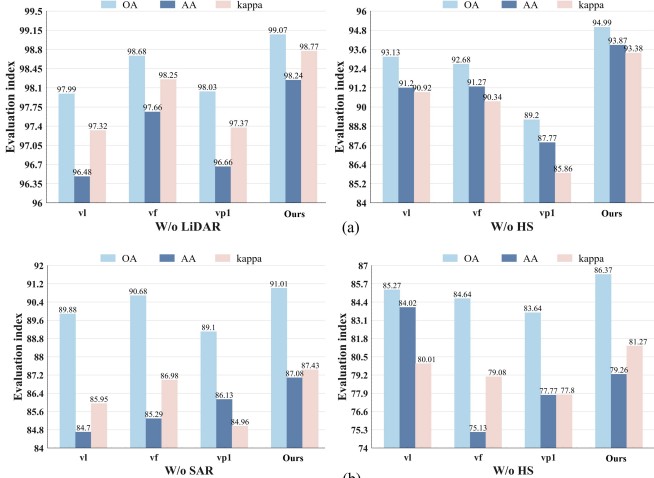

**Figure 7: Classification results of different variants for studying effectiveness of the LVPCnet on (a) Trento dataset, (b) Augsburg dataset.**

## 4.4 Ablation Study

*4.4.1 Effectiveness of Language Priors.* In order to investigate the effectiveness of language prior-driven visual feature extraction, we discuss a variant (named 'vl') of visual feature decoupling. This variant learns shared features by minimizing the Jensen-Shannon divergence between probability distributions of feature representations and employs domain classification objectives for specific feature learning. The comparative results are shown in Figure 7. After removing the text prior, OA of vl with the absence of HS images decreased by 1.86% and 1.1% compared to LVPCnet on the Trento and Augsburg datasets, respectively. This indicates that the introduction of text prior enhances visual representation learning, extracting more discriminative complementary information.

*4.4.2 Effectiveness of Feature Decoupling.* The proposed method captures modality-specific information while suppresses redundant

## 5 CONCLUSION

In this paper, LVPCnet is proposed to address the issue of modality missing in joint classification of multi-modal remote sensing images by compensating for specific features of the missing modality. The network is designed with a two-stage process for extracting specific complementary information from each modality and learning cross-modal specific information. This facilitates the recovery of specific features of the missing modality from known modalities when dealing with modality absence. Specifically, LVPCnet utilizes language priors to drive visual decomposition to explore complementary representations of multi-modal data, reducing redundancy. Subsequently, by embedding visual prompts, the model is guided to learn specific features of the missing modality from the known modalities, enabling the acquisition of complete multi-modal complementary information for joint classification. Systematic experimental investigations have been conducted on three public datasets to validate the effectiveness of our method.

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
