# OpenReview forum: "Language-Guided Visual Prompt Compensation for Multi-Modal Remote Sensing Image Classification with Modality Absence"
_acmmm.org/ACMMM/2024/Conference — MM2024 Poster_

### Official Review · Reviewer_AJPk · 2024-05-24

**Rating:** 5
**Confidence:** 3

**Summary:**

This paper proposes a language-guided visual prompt compensation network to achieve joint classification in case of arbitrary modality absence. The network consists of a language-guided visual feature decoupling stage (LVFD stage) and an absence-aware visual prompt compensation stage (VPC stage). LVFD-stage extract shared and specific modal features. VPC-stage learn visual prompts containing missing modality-specific knowledge through cross-modal representation alignment.

**Strengths:**

1. The article presents an innovative approach. It introduces the concept of utilizing language guidance to decompose modality information into shared and specific features, supplementing the specific information of the absent modality using visual prompts, which is a novel idea.
2. The article is well-structured, providing a comprehensive and explicit description of the research motivation and methodology design.
3. The proposed method exhibits strong performance. The unified network demonstrates the capability to handle the absence of any modality data relatively well compared to other methods.

**Limitations:**

1.	The paper lacks ablation experiments for each hyperparameter.

**Suitability:**

3

---

### Official Review · Reviewer_ZnhY · 2024-05-24

**Rating:** 4
**Confidence:** 3

**Summary:**

This paper proposed LVPCnet to address the issue of modality missing in the joint classification of multi-modal remote sensing images by compensating for specific features of the missing modality. The network is designed with a two-stage process for extracting specific complementary information from each modality and learning crossmodal specific information. Specifically, a language-guided visual feature decoupling stage (LVFD-stage) is designed to extract shared modality-specific features from multimodal images to build a complementary representation model of the complete modality. Subsequently, a missing perceptual visual cue compensation stage (VPC-stage) is proposed to learn visual cues containing missing modality-specific knowledge through cross-modal representation alignment, which further guides the complementary representation model to reconstruct modality-specific features of the missing modality from the available modalities based on the learned cues.

**Strengths:**

1, The method of using language-driven prompts proposed in this paper is innovative and inspiring and has achieved good results experimentally.

2, The paper is well organized and clear.

3, The two key stages are technically sound.

**Limitations:**

1, This paper needs to discuss the differences and connections between their approach and existing methods in the related work section, in addition, there is a lack of related work on multimodal remote sensing image classification in the task context.

2, Lack of ablation experiments for hyperparameters.

**Suitability:**

2

---

### Official Review · Reviewer_iG3q · 2024-05-24

**Rating:** 4
**Confidence:** 3

**Summary:**

The article proposes a language-guided visual prompt compensation network (LVPCnet) that uses a unified model while taking into account modality complementarity for joint classification in the presence of arbitrary modality absence. It embeds missing modality-specific knowledge into visual cues to guide the model in capturing complete modality information for classification from the available modality information.

**Strengths:**

1. a unified model LVPCnet for joint classification of arbitrary modal missing is proposed, which can cope with a wide range of modal missing situations and is highly practical.
2. the article explains the specific modules clearly, and explains the role of the modules superficially through diagrams and examples.
3. the article is very detailed in classifying and analyzing the existing methods, and summarizes the advantages and disadvantages of each class of methods

**Limitations:**

1. There is a lack of explanation of X^(m_i ) in Eq. (1), and a specific explanation of the meaning should be given at the first appearance of the symbol.
2. In Table 2, in the absence of modes, the LVPCnet method does not perform better than the traditional multimodal model as claimed in the text, e.g., in the comparisons in rows 2, 3, and 6, the AA metrics of BASELINE are better than those of LVPCNet.
3. The specific category names are not uniform, Figures 2 and 3 show Healthy grass, but in Figure 4 it is Health grass.

**Suitability:**

2

---

### Official Review · Reviewer_iyj8 · 2024-05-24

**Rating:** 4
**Confidence:** 3

**Summary:**

The paper titled "Language-Guided Visual Prompt Compensation for Multi-Modal Remote Sensing Image Classification with Modality Absence" presents a novel approach to address the challenge of modality absence in the joint classification of multi-modal remote sensing images. The authors propose a Language-Guided Visual Prompt Compensation Network (LVPCnet) that leverages language priors to guide the model in capturing complete modal information from available modalities for classification, even when some modalities are missing. The method involves a two-stage training process: a language-driven visual feature decoupling stage and an absence-aware visual prompt compensation stage. The effectiveness of the proposed approach is validated through systematic experiments on three public datasets.

**Strengths:**

Novelty: The paper introduces a unique solution to the problem of modality absence in multi-modal remote sensing image classification by utilizing language-guided visual prompts. This approach is innovative and differs from existing methods that either generate missing modalities or transfer knowledge from complete to incomplete modal sets.

Theoretical Approach: The theoretical foundation of using language priors to guide the feature extraction process is sound and aligns with current trends in multi-modal learning.

Technical Correctness: The proposed LVPCnet is technically well-defined, with a clear explanation of the language-driven visual feature decoupling and absence-aware visual prompt compensation stages.

Adequate Evaluation: The paper provides a comprehensive evaluation of the proposed method on three public datasets, comparing it with state-of-the-art methods, which demonstrates the effectiveness of LVPCnet.

Clarity: The paper is well-structured and clearly presents the methodology, experimental setup, and results. The use of figures and tables enhances the clarity of the presentation.

Applications: The proposed method has potential applications in various remote sensing tasks such as land analysis, urban planning, and environmental monitoring, where modality absence is a common issue.

**Limitations:**

Lack of Comparative Analysis: While the paper compares the proposed method with several existing approaches, it lacks a deeper comparative analysis that might reveal the trade-offs involved in using language-guided prompts versus other techniques.

Generalization: The paper does not provide extensive experiments to show how well the proposed method generalizes across different types of modality absence scenarios or different datasets.

Computational Efficiency: The paper does not discuss the computational efficiency of the proposed method, which is crucial for practical applications, especially in the context of remote sensing where data volumes are typically large.

Limited Discussion on Prompt Learning: The paper introduces the concept of language-guided visual prompts but does not delve into the nuances of prompt engineering, such as the selection, design, and optimization of these prompts for different missing modality scenarios.

**Suitability:**

2

---

### Meta-Review · Area_Chair_YdVm · 2024-07-02

**Recommendation:** Accept (Poster)
**Confidence:** 5

**Metareview:**

The paper got four acceptances after rebuttal. An acceptance is OK for this paper. I suggest the authors improve the writing and have more discussion according to the reviews and rebuttals.